# Cumulative Effect of Metabolic Factors on Hepatic Steatosis

**DOI:** 10.3390/diagnostics15182406

**Published:** 2025-09-22

**Authors:** Anna Egresi, Brigitta Kozma, Márton Karácsony, Aladár Rónaszéki, Klára Werling, Barbara Csongrády, Pál Kaposi Novák, Anikó Folhoffer, Attila Szijártó, Krisztina Hagymási

**Affiliations:** 1Department of Surgery, Transplantation and Gastroenterology, Semmelweis University, 1082 Budapest, Hungary; kozma.brigitta@semmelweis.hu (B.K.); karacsony.marci@gmail.com (M.K.); werling.klara@semmelweis.hu (K.W.); szijarto.attila@semmelweis.hu (A.S.); hagymasi.krisztina@semmelweis.hu (K.H.); 2Medical Imaging Center, Semmelweis University, 1082 Budapest, Hungary; ronaszeki.aladar@semmelweis.hu (A.R.); csongrady.barbara@semmelweis.hu (B.C.); kaposi.novak.pal@semmelweis.hu (P.K.N.); 3Department of Internal Medicine and Oncology, Semmelweis University, 1083 Budapest, Hungary; folhoffer.aniko@semmelweis.hu

**Keywords:** metabolic dysfunction-associated steatotic liver disease, obesity, type 2 diabetes (T2D), cardiometabolic risk

## Abstract

**Background/Objectives:** Hepatic steatosis, a hallmark of metabolic dysfunction-associated steatotic liver disease (MASLD), is closely associated with systemic metabolic dysfunction. However, the cumulative impact of metabolic risk factors on liver fat content remains underexplored. To evaluate the association between metabolic risk factors and hepatic steatosis severity using magnetic resonance imaging proton density fat fraction (MR-PDFF) measurement, and to assess the cumulative effect of multiple metabolic abnormalities. **Methods**: In this cross-sectional study, MASLD patients (*n* = 132, aged ≥ 18 years, age: 61.3 ± 10.3, male: 54, female: 78) underwent metabolic risk assessment and MR-PDFF liver fat content measurement. The association between certain metabolic risk scores (obesity/overweight, hypertension, hypercholesterolemia, hypertriglyceridemia, impaired fasting glucose or type 2 diabetes mellitus) both continuous and categorized, as well as liver fat content was analyzed using linear regression models. The cumulative effect of increasing metabolic risk was further explored with subgroup comparisons. **Results**: A significant positive association was observed between continuous metabolic risk scores and MR-PDFF values (β = 0.021, *p* < 0.001). Participants with higher cumulative metabolic risk (4 and 5 risk factors group) showed significantly higher liver fat content compared to the reference group (*p* < 0.01) (MetfO0 = 5.7 ± 5.9%; MetfO1 = 11.6 ± 9.5%; MetfO2 = 7.9 ± 5.6%; MetfO3 = 10.2 ± 7.9%; MetfO4 = 16.4 ± 11.0%; MetfO5 = 17.8 ± 9.5%). Intermediate metabolic risk categories showed a trend toward increased steatosis but did not reach statistical significance. **Conclusions**: Cumulative metabolic risk is strongly associated with increased hepatic fat accumulation. These findings underscore the need for early identification and management of metabolic risk factors to prevent the development and progression of hepatic steatosis.

## 1. Introduction

Metabolic dysfunction-associated steatotic liver disease (MASLD), formerly known as nonalcoholic fatty liver disease (NAFLD), has emerged as the most prevalent chronic liver condition worldwide, closely paralleling the rise in metabolic disorders such as obesity, type 2 diabetes mellitus, and cardiovascular disease (CVD).

With the global rise in obesity and type 2 diabetes, the number of individuals affected by MASLD is also climbing. At present, MASLD affects approximately 38% of adults and between 7% and 14% of children and adolescents. Projections suggest that by 2040, more than 55% of the adult population could be living with this condition. Although MASLD does not always advance to severe liver disease, it is now the leading cause of liver transplantation in the United States, especially among women and individuals with hepatocellular carcinoma (HCC) [1].

Recent research highlights a complex interplay between metabolic risk factors and hepatic steatosis, moving beyond the traditional “two-hit” hypothesis to a “multi-hit” model that integrates insulin resistance, adipose tissue dysfunction, gut microbiota alterations, and genetic predisposition as critical drivers of disease pathogenesis [2].

The relationship between metabolic syndrome (MetS) and MASLD is intricate and likely works in both directions. This interaction engages multiple organs—including the liver, pancreas, adipose tissue, and skeletal muscle—and is mediated by signaling pathways involving hepatokines. The result is a pro-inflammatory state that disrupts normal metabolism and heightens the risk of various metabolic disorders. Due to the strong link between obesity and metabolic risk factors, the ongoing global obesity epidemic is anticipated to further fuel this harmful cycle [1].

Insulin resistance and central obesity have been consistently identified as primary contributors to the development of hepatic steatosis. These metabolic abnormalities promote hepatic lipid accumulation through dysregulated lipolysis, increased free fatty acid flux to the liver, and de novo lipogenesis [2]. Furthermore, a growing body of evidence supports the role of genetic factors—particularly polymorphisms in genes such as PNPLA3 and TM6SF2—in modifying individual susceptibility to both liver fat accumulation and subsequent progression to fibrosis and cirrhosis [3].

Importantly, MASLD is not an isolated hepatic condition but is strongly linked with systemic metabolic dysfunction and adverse cardiovascular outcomes. This connection underscores the need for early identification of at-risk individuals and the implementation of holistic management strategies. Lifestyle interventions, including diet modification, regular physical activity, and smoking cessation, have been shown to significantly reduce the risk of MASLD, even among individuals with genetic predispositions [4].

Beyond its impact on liver health—such as the development of cirrhosis and HCC—MASLD is linked to a heightened risk of several non-liver conditions. These include the onset of type 2 diabetes, chronic kidney disease, sarcopenia, and various cancers outside the liver. Additionally, MASLD has been associated with a lower quality of life, reduced cognitive performance, fatigue, depression, diminished work productivity, and greater use of healthcare services, all contributing to a significant economic burden [1].

Given the rising global burden of metabolic syndrome and related liver disease, understanding the intricate relationship between metabolic risk factors and hepatic steatosis is critical. This study aims to further elucidate these associations by evaluating the relative impact of individual and combined metabolic risk factors on the presence and severity of hepatic steatosis in a contemporary adult population.

### Objectives

The primary objective of this study is to investigate the influence of key metabolic risk factors—including body weight, insulin resistance, dyslipidemia, and hypertension—on the development and severity of hepatic steatosis in adults.

Specifically, the study aims to evaluate the individual and combined effects of metabolic parameters on hepatic fat accumulation and to identify metabolic profiles that may predict an increased risk of advanced liver disease.

Through this multifaceted analysis, the study seeks to enhance current understanding of MASLD pathogenesis and support the development of more targeted prevention and management strategies.

## 2. Materials and Methods

### 2.1. Study Design and Population

This was a cross-sectional study conducted at Department of Surgery, Transplantation and Gastroenterology, Medical Imaging Center and Department of Internal Medicine and Oncology of Semmelweis University, Budapest. Adult participants (*n* = 132, aged ≥ 18 years, age: 61.3 ± 10.3, male: 54, female: 78) undergoing clinical evaluation for metabolic health were consecutively enrolled. Inclusion criteria included the availability of complete metabolic profiles and hepatic fat quantification by magnetic resonance imaging-proton density fat fraction (MR-PDFF). Exclusion criteria included significant alcohol consumption (defined as >20 g/day for women and >30 g/day for men), known viral hepatitis, autoimmune/cholestatic liver disease (primary biliary cholangitis, primary sclerosing cholangitis), storage disorders (Wilson’s disease, hemochromatosis) or use of hepatotoxic medications.

### 2.2. Clinical and Laboratory Assessments

Demographic data and medical history were collected. Anthropometric measurements, including body mass index (BMI), and blood pressure, were obtained by trained personnel.

Blood samples were collected following an overnight fast to measure glucose, lipid profiles (total cholesterol, LDL-C, HDL-C, triglycerides), and liver enzymes (ALT, AST).

#### Liver Fat Quantification

Hepatic steatosis was assessed using MR-PDFF performed on a 3T scanner All patients were scanned with a 1.5 T Philips IngeniaTM MRI scanner (Philips Healthcare, Amsterdam, The Netherlands) and a Q-Body coil. MR-PDFF values were used to classify steatosis severity into four categories:

Grade 0: <6%.

Grade 1: 6–17%.

Grade 2: 17–22%.

Grade 3: ≥22%.

### 2.3. Observed Metabolic Risk Factors

Obesity/overweight according to body mass index: normal weight (BMI: 18.5 kg/m^2^ to 24.9 kg/m^2^), overweight (BMI: 25.0 kg/m^2^ to 29.9 kg/m^2^), and obese (BMI: 30 kg/m^2^ or more).Hypercholesterolemia: total cholesterol > 5.2 mmol/L, or >200 mg/dL.Hypertriglyceridemia: triglycerides > 1.7 mmol/L, or >150 mg/dL.Impaired glucose metabolism: impaired fasting glucose ≥ 100 mg/dL or diagnosis of type 2 diabetes mellitus.Hypertension: blood pressure over 120/80 mmHg, or under treatment.

### 2.4. Cumulative Metabolic Risk Score

Each individual risk factor was scored as present or absent, and a cumulative metabolic risk score (range: 0–5) was calculated for each participant (MetfO0-MetfO1-MetfO2-MetfO3-MetfO4-MetfO5).

### 2.5. Statistical Analysis

Continuous variables were expressed as mean ± standard deviation (SD) or median (interquartile range, IQR) and compared using Fisher exact test, as appropriate. The association between individual and cumulative metabolic risk factors and steatosis grade (MR-PDFF categories) was assessed using multivariable ordinal logistic regression, adjusting for age and sex. A *p*-value < 0.05 was considered statistically significant. The regression models were adjusted for key confounders including BMI. Missing data were handled by case-wise exclusion. Assumptions of normality and homoscedasticity were tested and satisfied for all regression analyses. Statistical analyses were conducted using R (version 4.3.3. accessed on 29 February 2024).

## 3. Results

A total of 132 participants were included in the analysis. The mean hepatic fat fraction (MR-PDFF) was 6.13 ± 0.10%. The general lab parameters data presented in Table 1.

### 3.1. Association Between Metabolic Risk Factors and Hepatic Steatosis

Linear regression analysis demonstrated a significant positive association between the cumulative metabolic risk score (Metf) and hepatic fat fraction (Pdff) (β = 0.021, *p* = 0.0001). The model had an R^2^ of 0.1137, indicating that approximately 11% of the variance in hepatic fat fraction was explained by the cumulative metabolic risk factors (Table 2).

When evaluating the categorical effects of increasing metabolic burden (MetfO levels), a stronger association was observed. The model showed an improved R^2^ of 0.1757 (adjusted R^2^ = 0.1416), with a global model significance (*p* = 0.00025).

Specifically, compared to the reference group (MetfO0), higher categories MetfO4 and MetfO5 were independently associated with significantly increased hepatic fat content (MetfO4: β = 0.107, *p* = 0.0007; MetfO5: β = 0.122, *p* = 0.0008). (Table 3).

### 3.2. Post Hoc Comparisons

Tukey’s multiple comparisons revealed that participants who have 4 and 5 risk factors (MetfO4 and MetfO5) had significantly higher steatosis grade compared to those without metabolic risk factors (MetfO4 vs. MetfO0, *p* = 0.0084; MetfO5 vs. MetfO0, *p* = 0.0106). Significant differences were also found between MetfO2 and MetfO4 (*p* = 0.0057) and between MetfO2 and MetfO5 (*p* = 0.0132). (Table 4).

### 3.3. Adjusted Odds Ratios for Outcomes

Adjusted odds ratios (AORs) indicated that higher metabolic burden (4 metabolic risk factor) was significantly associated with increased odds of hepatic steatosis (AOR 0.3299, *p* = 0.0329). (Table 5, Figure 1 and Figure 2). However metabolic risk factors alone did not show significant association with hepatic steatosis. (Figure 2).

## 4. Discussion

Metabolic dysfunction-associated steatotic liver disease (MASLD) has emerged as the leading cause of chronic liver disease worldwide, with its incidence projected to increase further. It is closely associated with obesity, type 2 diabetes, and other metabolic risk conditions. MASLD contributes to a heightened risk of cardiovascular disease, chronic kidney disease, various cancers within and outside the liver, as well as liver-specific complications such as liver failure and hepatocellular carcinoma. Due to its extensive health and economic impact, MASLD represents a significant global public health issue that demands urgent attention from healthcare professionals and policymakers [1,2,3,4,5].

In this study, we evaluated the relationship between metabolic risk factors and hepatic steatosis measured by MR-PDFF, as a gold standard non-invasive imaging-based steatosis quantification method. Our findings demonstrated that an increasing number of metabolic abnormalities was significantly associated with higher degrees of hepatic steatosis. Notably, the cumulative metabolic risk score showed a strong linear association with liver fat content, supporting the concept that metabolic dysfunction plays a central role in hepatic fat accumulation.

The continuous metabolic risk score (Metf) was significantly correlated with MR-PDFF, indicating that even incremental changes in metabolic burden may influence liver fat deposition. Furthermore, when categorized, participants with higher levels of metabolic risk (4 and 5 risk factors) exhibited significantly higher liver fat fractions compared to the reference group, emphasizing a cumulative effect. This trend remained statistically significant after adjusting for potential confounders such as age, sex, and BMI. The metabolic risk factors alone did not show significant association with hepatic steatosis grade. However R^2^ values are modest, and is typical in biological research due to the multifactorial nature of MASLD.

These results align with previous studies suggesting that metabolic syndrome components—including insulin resistance, obesity, dyslipidemia, and hypertension—are major drivers of hepatic steatosis progression. The observed associations reinforce the emerging concept that MASLD is a hepatic manifestation of systemic metabolic dysfunction rather than an isolated liver disease [5,6,7].

The development of hepatic steatosis in the context of metabolic risk factors is driven by a complex interplay of hormonal, metabolic, and inflammatory mechanisms. Central to this process is insulin resistance, a common feature of obesity and type 2 diabetes mellitus, which disrupts normal lipid metabolism.

The National Health and Nutrition Examination Survey (NHANES) found that, compared to individuals who were metabolically healthy, those with type 2 diabetes (T2D) had an 11-fold higher likelihood of having MASLD. Additionally, individuals with prediabetes and those classified as metabolically unhealthy had approximately 4-fold and 3.4-fold increased risks for liver steatosis, respectively. The combination of MASLD and T2D was linked to the highest rates of both overall and cause-specific mortality, followed by those with prediabetes, metabolically unhealthy individuals without diabetes, and finally, metabolically healthy participants [8]. A recent meta-analysis found that between 2016 and 2021, 68.8% with type 2 diabetes (T2D) were affected by MASLD—reflecting a 13% increase compared to the 55.6% prevalence reported during the 1990–2004 period [2].

In insulin-resistant states, adipose tissue undergoes enhanced lipolysis, resulting in elevated circulating free fatty acids (FFAs). These FFAs are taken up by hepatocytes and esterified into triglycerides, leading to lipid accumulation within the liver [5,6,7].

In addition to insulin resistance, obesity contributes to hepatic steatosis through adipose tissue dysfunction. It is assumed that one in four individuals—nearly 2 billion people—could be living with obesity by 2035. The report also projected that childhood obesity may more than double by that time, with cases among boys rising by 100% to 208 million and among girls by 125% to 175 million [8]. Researchers found that the combined prevalence of MASLD was 70% among individuals who were overweight and 75% among those with obesity. Additionally, the rate of metabolic dysfunction-associated steatohepatitis (MASH), the more severe form of the disease, was 34% in both groups. When examining fibrosis severity, stages F2–F4—which are strongly linked to increased risk of chronic liver disease (CLD)-related mortality—were present in 20% of the overweight group and 22% of the obese group. Advanced fibrosis (stages F3–F4) was identified in 7% of individuals in both categories [1,9,10,11,12,13,14].

Enlarged adipocytes in visceral fat depots become inflamed and secrete pro-inflammatory cytokines, including tumor necrosis factor-alpha (TNF-α) and interleukin-6 (IL-6). These mediators exacerbate systemic and hepatic insulin resistance and directly promote hepatic inflammation. At the same time, levels of adiponectin, an adipokine with insulin-sensitizing and anti-steatotic properties, are typically reduced, further impairing hepatic lipid metabolism [6,7].

Dyslipidemia, commonly observed in individuals with metabolic syndrome, also plays a significant role in the pathogenesis of steatosis. Elevated levels of very-low-density lipoproteins (VLDL), low-density lipoproteins (LDL), and triglycerides contribute to increased hepatic lipid influx. Moreover, hepatic accumulation of cholesterol can induce lipotoxicity, activating inflammatory and fibrogenic pathways that may accelerate disease progression [8].

Hypertension is present in roughly 40% of individuals with MAFLD, and epidemiological data suggest that nearly half (49.5%) of people with high blood pressure also have MAFLD, highlighting a likely two-way relationship between the two conditions. Several mechanisms may contribute to the development of hypertension in those with MAFLD [15]. Notably, both the prevalence and incidence of hypertension rise progressively with advancing liver fibrosis, reaching their highest levels in individuals with stage F4 fibrosis, or cirrhosis [16].

Although hypertension does not directly induce steatosis, it is associated with endothelial dysfunction, oxidative stress, and hepatic hypoxia, which may potentiate liver injury and fibrogenesis in the context of metabolic liver disease. Furthermore, mitochondrial dysfunction, often triggered by excessive FFA load, results in incomplete fatty acid oxidation and the generation of reactive oxygen species (ROS). The resulting oxidative stress not only damages hepatocellular components but also amplifies inflammation and promotes fibrosis [8,9].

Lastly, alterations in the gut–liver axis, frequently observed in metabolic disorders, may contribute to hepatic steatosis through increased intestinal permeability and translocation of endotoxins such as lipopolysaccharides (LPS). These endotoxins activate toll-like receptor 4 (TLR4) signaling in Kupffer cells, promoting hepatic inflammation and further metabolic dysregulation [7].

Interestingly, intermediate risk groups (with 1, 2, and 3 risk, MetfO1, MetfO2, MetfO3) did not reach statistical significance in comparison to the lowest risk category (MetfOo). This might suggest a threshold effect, where mild to moderate metabolic disturbances do not markedly impact liver fat accumulation until a critical burden of metabolic abnormalities is surpassed.

The strengths of our study include the use of MR-PDFF, a non-invasive and highly accurate method for quantifying liver fat, and the assessment of both continuous and categorical metabolic risk burdens. However, limitations should be acknowledged. The cross-sectional design precludes causal inferences, and the relatively small sample size, particularly in certain categories, may have limited the statistical power to detect smaller differences.

Given the strong mechanistic and epidemiological links between metabolic risk factors and hepatic steatosis, there is a compelling rationale for identifying individuals at increased risk of progressive liver disease. Targeted screening of patients with established metabolic dysfunction—such as obesity, type 2 diabetes, insulin resistance, dyslipidemia, and hypertension—may offer a more efficient and clinically meaningful approach compared to universal, population-based screening strategies [10,11].

Patients with multiple metabolic risk factors are more likely to exhibit not only hepatic steatosis but also more advanced stages of liver injury, including steatohepatitis, fibrosis, and cirrhosis. Numerous studies have demonstrated that individuals with type 2 diabetes or obesity, particularly when accompanied by additional cardiometabolic abnormalities, have a significantly higher prevalence of advanced fibrosis and are at elevated risk for liver-related complications and mortality. Therefore, early identification of steatosis in this high-risk subgroup is essential to implement timely interventions that may halt or reverse disease progression [12]. Non-invasive imaging tools such as ultrasound and transient elastography, as well as biomarker-based indices (e.g., FLI, NAFLD fibrosis score), offer feasible options for screening steatosis and fibrosis in metabolically high-risk groups; however, in large-scale screening, their routine use in the general population may lead to overdiagnosis and resource strain without a clear benefit in outcomes [12,13].

In addition to liver-specific evaluations, it is essential to implement cardiovascular risk screening programs in high-risk populations, particularly those with metabolic dysfunction, because the most common cause of mortality among these patients is cardiovascular disease [1]. Although the precise biological mechanisms connecting MASLD and cardiovascular disease (CVD) are not yet fully understood, both conditions share overlapping metabolic disturbances that may act together to amplify the risk of disease progression in both organs. As a result, individuals may face a higher likelihood of coronary artery disease, left ventricular diastolic dysfunction and hypertrophy, calcification of heart valves, arrhythmias—particularly persistent atrial fibrillation—and heart failure [1,17,18,19,20].

Major professional societies, including the European Association for the Study of the Liver (EASL), the American Association for the Study of Liver Diseases (AASLD), the European Society of Cardiology (ESC), and the American Heart Association (AHA), consistently emphasize the importance of cardiovascular risk assessment in all patients diagnosed with MASLD. Although individuals with concomitant type 2 diabetes or advanced fibrosis represent a subgroup at particularly elevated risk, CVD remains the leading cause of morbidity and mortality in MASLD. Therefore, comprehensive cardiovascular risk assessment should be considered a standard component of care for all affected individuals. Several validated risk stratification tools—such as the Framingham Risk Score, SCORE2, and the ASCVD risk calculator—can be utilized to estimate cardiovascular risk and guide preventive strategies in this patient population [12,21,22,23,24,25,26].

Following established guidelines for the optimal management of cardiometabolic risks (CMRs) is strongly advised. This is particularly crucial because the CMRs linked to MASLD significantly overlap with those associated with cardiovascular disease—the leading cause of death in individuals with MASLD. Addressing these risks together represents a key strategy for reducing the overall impact of MASLD [1].

## 5. Conclusions

In conclusion, our findings highlight the importance of comprehensive metabolic risk assessment in individuals at risk for hepatic steatosis. Early identification and intervention targeting multiple metabolic pathways could be crucial in preventing or mitigating MAFLD progression. Cardiovascular risk assessment in high-risk steatotic patients must be emphasized.

## Figures and Tables

**Figure 1 diagnostics-15-02406-f001:**
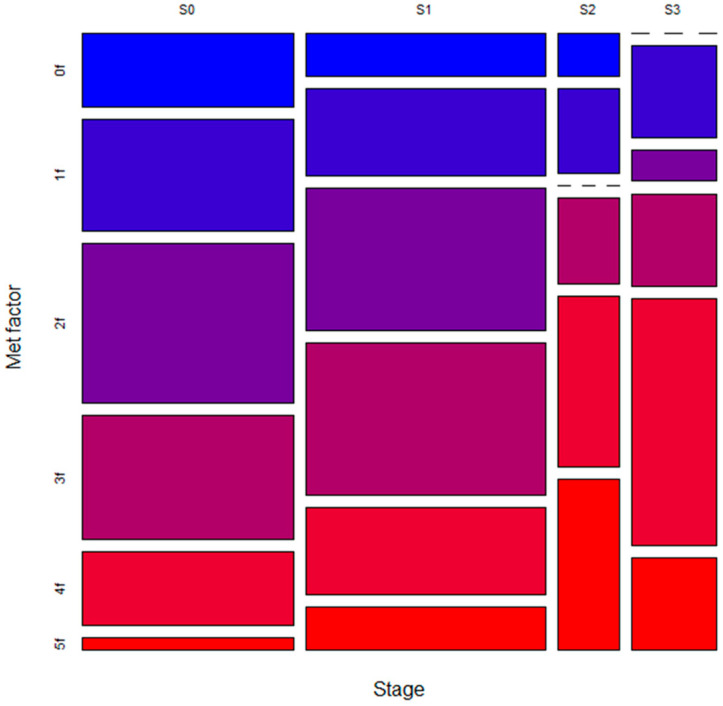
Mosaic plot displays the relationship between Metfactor (a categorical metabolic factor) and Stage (S0 to S3), which likely represents stages of liver disease progression, the severity of liver steatosis is indicated by color red.

**Figure 2 diagnostics-15-02406-f002:**
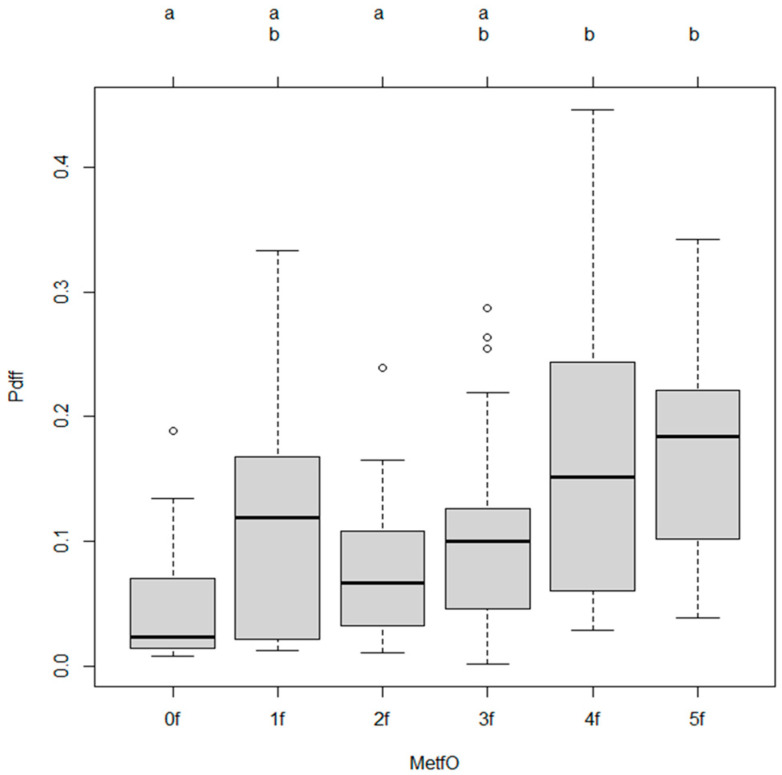
Association between hepatic steatosis and metabolic risk factors alone. This boxplot illustrates the distribution of metabolic burden (P.diff) across different levels of MetfO (Metabolic Factor Ordinal levels: 0f to 5f).

**Table 1 diagnostics-15-02406-t001:** General laboratory parameters of study population.

Parameter	Mean	Standard Deviation
BMI (kg/m^2^)	29.1	4.7
AST (U/L)	37.1	25.3
GPT (U/L)	47.7	38.2
GGT (U/L)	129	273.2
ALP (U/L)	93.8	47.5
Total bilirubin (umoL/L)	14.3	7.4
INR	1.1	0.6
Albumin (g/L)	43.9	5
Triglyceride (mmoL/L)	2.3	3
LDL-cholesterol (mmoL/L)	3.7	1.3
HDL-cholesterol (mmoL/L)	1.3	0.3
Thrombocyte (10^9^/L)	247	67
Glucose (mmoL/L)	5.9	1.6

**Table 2 diagnostics-15-02406-t002:** Association between Metabolic Risk Factors and Hepatic Steatosis (MR-PDFF).

Variable	β (Estimate)	Standard Error	*t*-Value	*p*-Value
Cumulative Metabolic Risk (Metf)	0.021	0.0053	4.005	0.0001
Intercept	0.0605	0.0158	3.838	0.0002

**Table 3 diagnostics-15-02406-t003:** Association by Metabolic Burden Categories (MetfO).

MetfO Category vs. Reference (0f)	β (Estimate)	Standard Error	*t*-Value	*p*-Value
MetfO1	0.0597	0.0315	1.896	0.0604
MetfO2f	0.0226	0.0305	0.742	0.4593
MetfO3f	0.0457	0.0302	1.514	0.1328
MetfO4f	0.1073	0.0307	3.500	0.0007
MetfO5f	0.1220	0.0356	3.427	0.0008

**Table 4 diagnostics-15-02406-t004:** Tukey post hoc Comparison (Selected Significant Results).

Comparison	Mean Difference	95% CI (Lower–Upper)	*p*-Value
MetfO4 vs. 0f	0.1073	0.0185–0.1961	0.0084
MetfO5 vs. 0f	0.1220	0.0189–0.2250	0.0106
MetfO4 vs. 2f	0.0847	0.0169–0.1525	0.0057
MetfO5 vs. 2f	0.0993	0.0137–0.1850	0.0132

**Table 5 diagnostics-15-02406-t005:** Adjusted odds ratios of metabolic risk factors on hepatic steatosis.

Variable	Beta Estimate	95% CI (Lower)	95% CI (Upper)	*p*-Value
Metf (continuous)	0.021	0.0107	0.0313	0.0001 ***
MetfO1f	0.0597	−0.0315	0.1509	0.0604
MetfO2f	0.0226	−0.0657	0.1109	0.4593
MetfO3f	0.0457	−0.0417	0.1331	0.1328
MetfO4f	0.1073	0.0185	0.1961	0.0007 ***
MetfO5f	0.1220	0.0189	0.2250	0.0008 ***

*** *p* < 0.001.

## Data Availability

The original contributions presented in this study are included in the article. Further inquiries can be directed to the corresponding author.

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
