# Peer review of "Cumulative Effect of Metabolic Factors on Hepatic Steatosis"

_diagnostics, 2025, doi:10.3390/diagnostics15182406_

Round 1

Reviewer 1 Report

Comments and Suggestions for Authors

The manuscript “Cumulative effect of metabolic factors on hepatic steatosis” by Anna Egresi et al. calculated the effect of different metabolic diseases on liver fat content.

Define  MR-PDFF in the Abstract when it is used for the first time.

Please rewrite this sentence for clarity “While 44 MASLD doesn't always progress to more severe liver disease, it has become the leading reason for liver transplantation in the United States, particularly among women and individuals diagnosed with hepatocellular carcinoma (HCC)”

Incomplete sentences “Identify potential metabolic profiles 90 that predict higher risk for advanced liver disease.” “Demographic information, medical history were collected“

Correct text formatting

“age:61,3±10,3,“ “HDL-cholezterin “please correct

“(PMID: 37227944)“ ?

“kg/m2“ please correct – uppercase

Please use identical format for both numbers 6.13 ± 0.1030

Please explain “G/L „

Table 2, and table 4 the column Significance should be deleted.

Figures have to be improved; the labels are too small, and a better description has to be provided in the figure legends.

The first paragraph of the Discussion should be deleted.

“The metabolic risk factors alone did not show a significant association with hepatic steatosis grade.” Does this indicate that BMI was not associated with liver fat?

“Additionally, individuals with prediabetes and those classified as metabolically unhealthy had approximately 4-fold and 3.4-fold increased risks, respectively” for MASLD?

“A recent meta-analysis examining the prevalence of MASLD in individuals with 240 type 2 diabetes” are these Europeans?

The discussion has to be rewritten. The discussion should focus on the study's findings.

The discussion is far too long.

Reviewer 2 Report

Comments and Suggestions for Authors
  1. Summary and Overall Impression

This manuscript presents a well-executed cross-sectional study investigating the cumulative effects of metabolic risk factors on hepatic steatosis, quantified using MR-PDFF. The authors aim to provide insights into how clustering metabolic abnormalities may influence liver fat accumulation, contributing to the growing understanding of MASLD pathogenesis.

The study is timely, relevant, and supported by robust imaging methods and statistical analyses. The manuscript is mostly well-written and structured, though some issues regarding clarity, methodology explanation, and discussion depth warrant attention.

 Major Comments

  1. Study Design and Methodology
  • Inclusion/exclusion criteria are appropriate, and the imaging modality used (MR-PDFF) is commendable for accuracy, but the MR protocol description is excessively technical for a general audience; consider summarizing only key acquisition parameters and moving full details to supplementary materials.
  • It will be helpful if justification is provided for the selected cut-offs for metabolic risk factors (e.g., why 120/80 mmHg for hypertension?).
  • There is limited detail on how missing data were handled (if any), and whether statistical assumptions (normality, homoscedasticity) were tested.
  1. Statistical Analysis
  • The R² values reported (e.g., 0.1137 and 0.1757) are quite low. While this is common in biological studies, it should be acknowledged and discussed as a limitation.
  • Please specify whether the regression models adjusted for confounders such as BMI, which may mediate associations between risk factors and liver fat.
  1. Results Presentation
  • Ensure figures are described with figure legends.
  • Results state both that individual metabolic factors were not significantly associated with steatosis and that obesity, T2D, and dyslipidemia have strong links. This contradiction should be explained.
  1. Discussion and Interpretation
  • Several paragraphs in the Discussion repeat background material (e.g., definitions, global burden data) that should be minimized.
  • The threshold effect in metabolic burden (i.e., ≥4 risk factors) is mentioned but not deeply explored, are these findings consistent with existing literature? Could a non-linear model better capture this relationship?
  • No discussion is provided on the implications for clinical screening or stratification beyond general lifestyle recommendations. Consider proposing how the cumulative risk score might be used in clinical practice or future research.
  1. References and Literature Context
  • Some in-text citations are missing journal names or formatted inconsistently (e.g., "[insert DOI]" in reference #14). Please ensure all references are finalized.

 Minor Comments

Section

Comment

Abstract

 Line 18: Typo in “hyertriglyceridaemia” → should be “hypertriglyceridemia”.

Introduction

 Clarify "multi-hit model" briefly for a general audience.

Methods

 Clarify if inter-rater variability in MR-PDFF readings was assessed.

Results

 Line 150: Hepatic fat fraction is listed as “6.13 ± 0.1030%” — verify this SD.    Seems implausibly small.

Tables

Consider condensing or moving some statistical tables to supplementary files.

Language

Proofreading required to fix typographical errors

Ethics

Well-reported and in accordance with the Declaration of Helsinki.

Round 2

Reviewer 1 Report

Comments and Suggestions for Authors

HDL-cholesterin, I think it is cholesterol 

yaged, please correct 

Table 3, the row "significance" should be deleted. 

Colours in figure 1 need explanation

Author Response

  • HDL-cholesterin, I think it is cholesterol.”
    Thank you for pointing this out. We have corrected the terminology to “HDL-cholesterol” throughout the manuscript to maintain consistency and accuracy.

  • “yaged, please correct.”
    The typographical error “yaged” has been corrected to the appropriate aged in the revised manuscript.

  • “Table 3, the row ‘significance’ should be deleted.”
    Thank you for this suggestion. We have removed the “significance” row from Table 3 as recommended.

  • “Colours in figure 1 need explanation.”
    We agree that clarification was needed. In the revised manuscript, we have added an explanation of the color coding in Figure 1.

Reviewer 2 Report

Comments and Suggestions for Authors

Accept the revised manuscript

Author Response

Thank you!